# Comprehensive Transcriptome Analysis Reveals Insights into Phylogeny and Positively Selected Genes of *Sillago* Species

**DOI:** 10.3390/ani10040633

**Published:** 2020-04-07

**Authors:** Fangrui Lou, Yuan Zhang, Na Song, Dongping Ji, Tianxiang Gao

**Affiliations:** 1Fishery College, Zhejiang Ocean University, Zhoushan 316022, Zhejiang, China; lfr199202@163.com; 2Fishery College, Ocean University of China, Qingdao 266003, Shandong, China; zhangyuan_ouc@163.com (Y.Z.); songna624@163.com (N.S.); 3Agricultural Machinery Service Center, Fangchenggang 538000, Guangxi, China; jidongping201@163.com

**Keywords:** Sillaginidae, bottom dweller, RNA-seq, orthologous exon markers, positive selection

## Abstract

**Simple Summary:**

A comprehensive transcriptome analysis revealed the phylogeny of seven *Sillago* species. Selection force analysis in seven *Sillago* species detected 44 genes positively selected relative to other Perciform fishes. The results of the present study can be used as a reference for the further adaptive evolution study of *Sillago* species.

**Abstract:**

*Sillago* species lives in the demersal environments and face multiple stressors, such as localized oxygen depletion, sulfide accumulation, and high turbidity. In this study, we performed transcriptome analyses of seven *Sillago* species to provide insights into the phylogeny and positively selected genes of this species. After de novo assembly, 82,024, 58,102, 63,807, 85,990, 102,185, 69,748, and 102,903 unigenes were generated from *S.*
*japonica*, *S.*
*aeolus*, *S.* sp.1, *S.*
*sihama*, *S.* sp.2, *S. parvisquamis*, and *S.*
*sinica*, respectively. Furthermore, 140 shared orthologous exon markers were identified and then applied to reconstruct the phylogenetic relationships of the seven *Sillago* species. The reconstructed phylogenetic structure was significantly congruent with the prevailing morphological and molecular biological view of *Sillago* species relationships. In addition, a total of 44 genes were identified to be positively selected, and these genes were potential participants in the stress response, material (carbohydrate, amino acid and lipid) and energy metabolism, growth and differentiation, embryogenesis, visual sense, and other biological processes. We suspected that these genes possibly allowed *Sillago* species to increase their ecological adaptation to multiple environmental stressors.

## 1. Introduction

The fish family *Sillago* are commonly known as sand whitings or sand borers, which are widely distributed in inshore waters of the Indian Ocean and the western Pacific Ocean and thrive in estuaries and shoals [1]. As minitype bottom-dwelling fishes of shallow sea regions, *Sillago* species are gregarious and have the ecological habit of drilling sand [2]. In addition, the *Sillago* species have become an important economical edible fish due to their palatable meat and long chain fatty acids that prevent thrombosis [2]. The inshore fishing of *Sillago* species has also developed rapidly in the past decades. It is worth noting that the overfishing will eventually lead to the ecological and economic damage of the *Sillago* resource. Additionally, more complex demersal environments caused by climate change and human activities, such as localized oxygen depletion, sulfide accumulation, and high turbidity, also pose a challenge to *Sillago* species differentiation and survival [3]. Therefore, it is necessary to develop the management and restoration of *Sillago* fishery resources.

Inferring the phylogenetic relationships is fundamental to successfully managing and recovering *Sillago* fishery resources, allowing the identification of the origin and relationship of species and the revelation of the evolutionary processes that lead to species diversity. At present, eleven *Sillago* species have been recorded in China [4]. Although there exist few taxa in *Sillago*, species identification and phylogenetic status are often confusing to taxonomists due to their similar phenotypic and physiological characteristics [2]. Additionally, local environmental divergences and rapid climatic changes could lead to the further diversification and speciation of *Sillago*, ultimately improving the complexity of phylogenetic research of *Sillago* species [5]. Although significant development has been reported in recent years, the knowledge of the phylogeny of *Sillago* species is still controversial, especially in the process of identifying cryptic lineages within the vast diversity of *Sillago* through the analysis of different phenotypic characteristics. Particular points of contention include (i) the incorrect use of species’ scientific names and (ii) species identification errors from identification based only on morphological features [1,2,4]. The application of comprehensive molecular phylogenies upended classical morphological hypotheses and led to the development of a new *Sillago* phylogenetic classification method. For example, four *Sillago* species, including *Sillago caudicula* [2], *S. sinica* [6], *S. suezensis* [7], and *S. shaoi* [4] were reclassified using molecular markers, although they were all wrongly classified as *S. sihama* based on phenotypic traits. It is impossible to construct a comprehensive phylogenetic relationship for *Sillago* species only by using a single or small number of gene fragments. This is the case because a small number of gene fragments contain limited genetic information, and therefore cannot provide high statistical support for some crucial branches of the phylogenetic tree [8]. Different gene fragments are also affected by sequence conservation, gene evolution rate differences, horizontal gene transfer, and other factors, which ultimately lead to conflicting gene trees [9]. Additionally, it is undeniable that all previous studies only analyzed the differentiation between *Sillago* species, while the genetically adaptive characteristics were still not clear. Therefore, it is very essential to investigate the molecular features of different *Sillago* species based on complete genetic information, which will provide us with an accurate understanding of differentiation and environmental adaptation in *Sillago* species. 

The sequence differentiation of orthologous genes is related to speciation, and therefore, orthologous genes can be used to infer more accurate phylogenetic relationships among species [10]. Currently, the advent and increasingly widespread use of high-throughput sequencing technologies provides hundreds to thousands of orthologous genes for the phylogenetic analysis of species [11,12]. However, it is worth noting that the massive datasets generated by high-throughput sequencing, especially whole genome datasets, require more complex analytical methods to confirm the most informative orthologous loci and appropriate tree reasoning methods [13]. For vertebrates (including teleost), two whole-genome duplications (WGDs) eventually produced duplicate genes, which further affected the accuracy of distinguishing orthologous genes from duplicated paralogs [14,15]. In *Sillago* species, unfortunately, there is only the *S. sinica* genome project that has been sequenced [5]. Modern transcriptome datasets may avoid the defects mentioned above, because they can effortlessly generate massive genome-wide protein-coding sequences for phylogenetic studies, especially when whole genomic information is not available [10]. Meanwhile, we can focus on calculating and comparing the gene evolutionary rates based on orthologous gene datasets, and then infer the potential local environmental adaptation genes of organisms [16,17]. In brief, we believe that the phylogenetic relationships and genetic adaptation characteristics will be successively identified in *Sillago* species by orthologous gene analysis.

In order to reconstruct the complete phylogenetic relationship of *Sillago* species and detect their genomic characters associated with adaptive evolution, five valid *Sillago* species (including *S. japonica*, *S. aeolus*, *S. sihama*, *S. parvisquamis*, and *S. sinica*) and two unpublished new species (*S.* sp.1 and *S.* sp.2; unpublished data)—which were confirmed by using morphological, anatomical, and DNA-barcoding evidence—were studied. Next, seven RNA sequencing (RNA-seq) libraries were constructed together. First, clean reads produced by RNA-seq were applied to assemble seven relatively integrated transcriptomes of *Sillago* species. Subsequently, transcriptome data of seven *Sillago* species and other existing Perciformes species were detected for orthologous genes, and then the shared orthologous genes were used to reconstruct the phylogenetic tree and predict the differentiation time. Considering that these genes that experience positive selection may be beneficial to improving the ecological adaptation of *Sillago* species, we investigated the positively selected genes (PSGs) of *Sillago* species based on orthologous genes. Meanwhile, a gene function enrichment analysis was performed on these PSGs. These results can constitute important data with which to gain insights into the process of species differentiation and adaptive evolution in other *Sillago* species.

## 2. Materials and Methods 

### 2.1. Ethics Approval and Consent to Participate

The *Sillago* species are not endangered or protected species in China or other countries. In addition, frost anesthesia was used to minimize suffering in all samples.

### 2.2. Sample Collection, RNA Extraction, and Illumina Sequencing 

Seven *Sillago* species were obtained from the coast of China and Japan. Then, one adult female per species was immediately euthanized, and the muscle of each individual was rapidly sampled, snap-frozen in liquid nitrogen, and stored at −80 °C prior to the RNA extraction. The sampling location and the standard length (SL) of each sequenced individual are shown in Figure 1. The total RNA of the seven *Sillago* species was extracted using a standard TRIzol Reagent kit, following the manufacturer’s protocol. The quantitative evaluation of total RNA was conducted by using the Agilent 2100 Bioanalyzer (Agilent Technologies, Santa Clara, CA, USA). We then purified mRNA by depleting rRNA from total RNA, and the remaining mRNA was cleaned three times. Then, we fragmented the mRNA into fragments of appropriate size. The fragmented mRNA was used to construct a cDNA library. Afterwards, we added A-tails and adapters to the double stranded cDNA. Then, the cDNA libraries were diluted to 10 pM and then quantified by using the Agilent 2100 Bioanalyzer. Finally, each tagged cDNA library was sequenced on the Illumina HiSeq 2000 across one lane with paired-end 150 bp reads.

### 2.3. Transcriptome De Novo Assembly

The raw RNA-seq data of *Sillago japonica*, *Sillago aeolus*, *Sillago sihama*, *Sillago parvisquamis*, *Sillago sinica*, *Sillago* sp.1 and *Sillago* sp.2 reported in this paper have been deposited in the Sequence Read Archive (SRA) database of National Center for Biotechnology Information (NCBI) under BioProject number PRJNA596307, with accession number of SRR10743284, SRR10743285, SRR10743281, SRR10743280, SRR10743279, SRR10743282 and SRR10743283, respectively. The FastQC 0.11.2 software (version, Babraham Institute, Cambridgeshire, UK) was applied to assess the sequencing quality of all raw reads in FASTQ format. Then, clean reads were obtained by removing reads with sequencing adapters, unknown nucleotides (N ratio > 10%), and low quality (quality scores < 20) using Trimmomatic 0.36 [18]. Additionally, the *S. sinica* genome dataset served as a reference sequence and was used for subsequent de novo assembly [5]. All remaining high-quality clean reads were sorted according to *S. sinica* genome index sequences and were aligned to the reference assembly generated from paired-end reads by using the bwa-mem algorithm in the BWA (Version 0.7.12, Microsoft Corporation, Redmond, WA, USA; [19]) software with default parameters, with the resulting output as ‘sam’ files. All sam files of each species were merged into a ‘bam’ file using the SAMtools (Version 1.3.1; Microsoft Corporation, Redmond, WA, USA; [20]) software. Finally, the Trinity (Version 2.4.0, Microsoft Corporation, Redmond, WA, USA; [21]) software was used for the transcriptome de novo assembly of each species, with the parameters as follows: --genome_guided_max_intron 10000. In order to perform further quantitative assessment of the assembly completeness, we applied the BUSCO software (version 4.0, Microsoft Corporation, Redmond, WA, USA) package with default settings, and downloaded the Ensembl Actinopterygii assembly as a reference.

### 2.4. Orthology Determination and Phylogenetic Tree Reconstruction

In order to reconstruct a more accurate phylogenetic tree, we performed an extensive orthologous gene comparison among the seven *Sillago* species and thirteen other Perciformes species (hereafter referred to as the “research species”) with transcriptome or genome datasets, including *Epinephelus fuscoguttatus* (GCNQ00000000.1), *Epinephelus coioides* (GCA_900536245.1), *Perca fluviatilis* (GCA_003412525.1), *Chionodraco hamatus* (GCA_009756495.1), *Gymnodraco acuticeps* (GGFR00000000.1), *Dissostichus eleginoides* (GHKE00000000.1), *Trematomus bernacchii* (GBXS00000000.1), *Argyrosomus regius* (GFVG00000000.1), *Nibea albiflora* (GCA_900327885.1), *Miichthys miiuy* (GCA_001593715.1), *Collichthys lucidus* (GCA_004119915.1), *Larimichthys polyactis* (GETG00000000.1), and *Larimichthys crocea* (GCA_000972845.2). Meanwhile, the ecological characteristics of the 20 research species were obtained from the Fishbase website [22] and the research results of Xiao et al. (2016) [4], as shown in Table 1. Firstly, we obtained the assembly sequences from the National Center for Biotechnology Information (NCBI) database under the accession number. Additionally, a dataset of 1721 single-copy conserved nuclear coding sequences (> 200bp) was obtained by comparing and optimizing eight well-annotated model fish genomes: *Lepisosteus oculatus*, *Anguilla anguilla*, *Danio rerio*, *Gadus morhua*, *Oryzias latipes*, *Oreochromis niloticus*, *Gasterosteus aculeatus*, and *Tetraodon nigroviridis* [23]. Subsequently, we extracted the single-copy conserved nuclear coding sequences of each research species using the HMMER software (Version 3.1, Howard Hughes Medical Institute, Chevy Chase, MD, USA; [24]). Specifically, 1721 parameterized hidden Markov model (HMM) profiles were obtained based on 1721 sequence alignments using the ‘hmmbuild’ function of the HMMER software. Each HMM model was applied to search against each research species dataset using the nHMMER program within HMMER, with the resulting hits output as a table. We used Python scripts (available on Dryad; [23]) to retain the hits that were at least 70% as long as the shortest model fragment and had a bit score of at least 100, and de-redundancy was performed on these hits with 100% similarity by using the CD-HIT software [25]. The remaining high-quality single-copy conserved nuclear coding sequences of each research species were aligned and spliced into single nucleotide sequences using the MAFFT software [26]. Conserved sequences were extracted from each concatenated nucleotide sequence using Gblocks with parameter -t=c [27]. Finally, we performed 1000 nonparametric bootstrap replicates for the optimal GTRGAMMA substitution model of all concatenated nucleotide sequences in the MEGA (Version 6.0, National Institutes of Health, Bethesda, MD, USA; [28]) software, and then a complete neighbor-joining (NJ) tree of the abovementioned twenty species was reconstructed according to the branch lengths and bootstrap support values. Additionally, we also reconstructed the NJ trees based on the amino acid sequences and variation sites of concatenated nucleotide sequences. The iTol (Version 4.0, Beijing Institute of Genomics, Beijing, China; [29]) software was used to visualize the phylogenetic relationship. Finally, the estimated molecular clock data of *L. polyactis* and *L. crocea* (min and max differentiation times are 11.2 and 42.0 MYA) were obtained from the TimeTree database [30], and then the divergence times of the seven *Sillago* species were estimated using the r8s software [31].

### 2.5. Prediction of PSGs of Sillago Species

In the present study, the codeml program in the PAML (Version 4.9, University College London, London, UK; [32]) software was applied to identify the PSGs of *Sillago* species. Firstly, we constructed tree files for 20 research species (including the seven Sillago species and 13 outgroups) using each single-copy orthologous gene, respectively. Subsequently, the branch-site model (model = 2, Nsites = 2) in the codeml program was used to identify the PSGs of *Sillago* species, and a comparison was also conducted between the null and alternative models. The null model assumed that *Sillago* species were under purifying selection and that therefore those sites on the foreground branch evolved neutrally (non-synonymous (dN)/synonymous (dS) = 1, modelA1, fix_omega = 1, and omega = 1.5), and the alternative model assumed that those sites on the foreground branch were under positive selection (dN/dS > 1, modelA2, fix_omega = 0, and omega = 1.5). Then, a likelihood ratio test (LRT) was applied to calculate the log-likelihood values (2△ln) between the null model and alternative model of each single-copy orthologous gene. After a Chi-square statistical analysis, a gene was considered as a PSG of *Sillago* species if the FDR-adjusted *p* < 0.01. Finally, we used the Blast2GO [33] software to predict the functions of those PSGs.

## 3. Results

### 3.1. Illumina Sequencing and the De Novo Assembly of the Seven Sillago Species’ Transcriptomes

Illumina sequencing was carried out on muscle from the seven *Sillago* species. After cleaning and quality testing, we obtained 138.4 Gb of clean reads from the seven *Sillago* species, and the details of the data are listed in Table 2. All clean reads of the present study were uploaded to the SRA databases of NCBI under BioProject number PRJNA596307, with accession numbers of SRR10743279 to SRR10743285. The Trinity software was applied to the de novo assembly of the clean data, and 82,024, 58,102, 102,185, 69,748, 102,903, 63,807, and 85,990 unigenes were generated from *S. aeolus*, *S. japonica*, *S. parvisquamis*, *S. sihama*, *S. sinica*, *S.* sp.1, and *S.* sp.2, respectively. The assembly information of the seven *Sillago* species is shown in Table 3. Additionally, the BUSCO analysis results showed that 91.5%, 83.1%, 94.2%, 91.7%, 93.1%, 88.7%, and 89.1% of protein-coding genes were found in the unigenes of *S. aeolus*, *S. japonica*, *S. parvisquamis*, *S. sihama*, *S. sinica*, *S.* sp.1, and *S.* sp.2, respectively.

### 3.2. Orthologous Gene Identification and the Phylogenetic Structure of Sillago Species

Single-copy conserved nuclear coding sequences of these eight model species were applied to search for orthologous genes in 20 research species. Using the HMMER software, we identified a set of 140 orthologous exon markers longer than 200bp. Then, the concatenated alignment of 140 orthologous genes produced a data matrix with 82,833 bp for 20 research species. The NJ analyses of 20 concatenated nucleotide sequences, concatenated amino acid sequences, and concatenated variation sites were implemented in the MEGA package, and the phylogenetic structures of *Sillago* species are shown in Figure 2, Figure 3 and Figure 4. According to all phylogenetic analyses, fishes from the same family eventually cluster into one branch, except for *C. hamatus* and *G. acuticeps*. Seven *Sillago* species, including two suspected new species, were also clearly distinguished from other Perciformes. Additionally, the divergence time was estimated, and the results showed that the internal divergence time varied within the seven *Sillago* species due to the different kind of concatenated sequence used for analysis. Specifically, the internal divergence times of the seven *Sillago* species based on concatenated nucleotide sequences was between 161.29 and 27.80 MYA, and the internal divergence time based on the amino acid sequences was between 27.02 and 4.15 MYA, while that based on variation sites was between 24.13 and 3.46 MYA.

### 3.3. PSGs Representative of Sillago Species

We calculated the log-likelihood values (2△ln) between the null model and alternative model for 140 orthologous genes. After a Chi-square statistical analysis, a total of 44 genes with FDR-adjusted *p* < 0.05 were identified as PSGs and these genes were suspected to contribute to the specific adaptive evolution of *Sillago* fishes in the burrowing lifestyle. (Table 4). Combining nr (non-redundant protein sequence) database annotation information, we speculate that these adaptive genes are potentially related to the stress response, material (amino acid and lipid) and energy metabolism, growth and differentiation, embryogenesis, visual sense, and other things. Further gene function enrichment analysis indicated that these adaptive genes were involved in cellular process (GO: 0050794), metabolic process (GO: 0006006), cell parts (GO: 0044464), cells (GO: 0005623), catalytic activity (GO: 0003824), binding (GO: 0005488), and other things (Figure 5). Additionally, we identified the networks of molecular interactions in the cells and variants specific to particular organisms by comparing the adaptive genes to the KEGG database (Table 5). Results showed that these adaptive genes were significantly enriched for nicotinate and nicotinamide metabolism (map00760), carbon fixation pathways in prokaryotes (map00720), purine metabolism (map00230), C5-Branched dibasic acid metabolism (map00660), starch and sucrose metabolism (map00500), arginine biosynthesis (map00220), riboflavin metabolism (map00740), pantothenate and CoA biosynthesis (map00770), pyrimidine metabolism (map00240), the biosynthesis of antibiotics (map01130), the citrate cycle (map00020), propanoate metabolism (map00640), thiamine metabolism (map00730), and arginine and proline metabolism (map00330).

## 4. Discussion

As they are typical marine demersal fish, species differentiation of *Sillago* species is often disordered, which ultimately affects the accurate interpretation by taxonomists of their evolutionary processes. Meanwhile, more complex habitat environments caused by climate change and human activities can also have an impact on the survival of *Sillago* species. However, it is undeniable that unraveling their complete phylogenetic relationships and adaptive mechanisms can effectively solve the above problems. In order to gather this fundamental knowledge, we first sequenced and assembled the transcriptome of seven *Sillago* species. Then, we identified the orthologous genes of seven *Sillago* species and 13 other Perciformes species based on transcriptome data. Finally, the phylogenetic relationships were reconstructed and the PSGs of *Sillago* species were inferred based on orthologous genes. In brief, we believe that this research can provide new perspectives for protecting the *Sillago* fishery resource.

### 4.1. Transcriptome Data Processing 

In the present study, we obtained 138.4 Gb of clean transcriptomic data from seven *Sillago* species. To our knowledge, this study may be the first systematic research of seven *Sillago* species’ transcriptomes by using high-throughput sequencing technology, except for *S. japonica* [34]. It is undeniable that the transcriptome assembly results of *S. japonica*, *S.* sp.1, and *S.* sp.2 are imperfect because their N50 lengths are less than 1000 bp. Meanwhile, the BUSCO results also found that the unigenes of *S. japonica*, *S.* sp.1, and *S.* sp.2 lacked integrity. This is the case because some of the reads contaminated by the sequencing process were removed. Additionally, the selection of the reference genome (the *S. sinica* genome was selected as the reference sequence in this study) and assembly strategy (software and parameters) can also influence the final assembly efficiency. However, we still believe that these data expand the currently available genomic resources for *Sillago* species.

### 4.2. More Accurately Determining the Phylogenetic Relationships of Seven Sillago Species

Knowledge of the phylogeny of *Sillago* species is insufficient due to their similar phenotypic and physiological characteristics. Additionally, the paucity of genomic resources has restricted the phylogenetic resolution of *Sillago* species relationships. Currently established high-throughput sequencing technologies enable systematists to acquire huge amounts of orthologous genes for phylogenetic relationship reconstruction for *Sillago* species [23]. It is well known that the differentiation of orthologous genes usually leads to the speciation of species [35]. Therefore, there is every reason to believe that we can more accurately determine the phylogenetic relationships of *Sillago* species based on hundreds to thousands of orthologous genes. Although more complete orthologous genes could be obtained based on whole-genome sequencing technology, their application in phylogenetic analysis is limited by the high sequencing cost and methodological challenges of big databases [13]. With this background, orthologous exon markers captured by transcriptome sequencing have been attempted to be used in phylogenetic studies [36]. In fact, Betancur-R et al. (2013) considered that exons can be translated to amino acids, and further, to reduce errors from base compositional biases in phylogenetic studies [37]. Therefore, we focused on orthologous exon markers to reconstruct the phylogenetic relationships between seven *Sillago* species in this study. Unfortunately, only 140 orthologous exon markers were obtained using the HMMER method, and we suspected it may be related to the imperfect assembly results for *S. japonica*, *S.* sp.1, and *S.* sp.2. Although the number of markers is small, their efficiency has been verified by Hughes et al. [23], thus we believed that these markers are suitable for subsequent phylogenetic relationship reconstruction. Unsurprisingly, the reconstructed phylogenetic structure based on 140 orthologous exon markers (nucleotide sequences, amino acid sequences, and variation sites) is significantly congruent with the prevailing morphological and molecular biological view of Perciformes species relationships, except for *C. hamatus* and *G. acuticeps*. In other words, fishes from the same family eventually cluster into one branch. The reconstructed phylogenetic relationships of seven *Sillago* species are consistent with the results based on the mitochondrial genome [38]. Our study also provided evidence to prove that *S.* sp.1 and *S.* sp.2 may belong to the cryptic species of *Sillago*. However, more detailed evidence, including ecological and morphological data, needs to be provided in the further definition of *S.* sp.1 and *S.* sp.2. Additionally, there is no denying the fact that we still need to supplement other *Sillago* transcriptome data to reconstruct a more comprehensive understanding of phylogenetic relationships. It is worth noting that *C. hamatus* and *G. acuticeps* belong to the family Channichthyidae and Bathydraconidae, respectively, but the two species were clustered into one branch in this study. We suspected that the accuracy of the data may have contributed to the divergence. Additionally, some markers that were used to construct phylogenetic relationships between *C. hamatus* and *G. acuticeps* may have genetic convergence [39], which eventually cause the two species to cluster into one branch. This also confirmed that the selection of genetic markers may influence the reconstruction results of phylogenetic relationships [40]. Further studies will need larger datasets to illustrate the divergence. 

The evolutionary sequence of *Sillago* species was identified based on differentiation times. The evolutional sequence of the seven *Sillago* species was consistent with that in previous studies [38]. Previous studies suspected that incomplete or missing swim bladders may be an adaptive mechanism for Sillaginidae species to demersal life [2]. In addition, Xiao (2018) also deduced that the swim bladders of Sillaginidae ancestors were extremely simple, and then became more complex as they evolved [38]. It is worth noting that the swim bladder of *S. aeolus* is imperfect relative to that in the other six *Sillago* species. Therefore, the evolutionary sequence of seven *Sillago* species based on transcriptome data also seems to support the hypothesis about the evolution of swim bladders. However, when evaluating the differentiation time of *Sillago* species based on different sequence formats, the results are quite different. In fact, Schwarzhans considered that Sillaginidae species gradually evolved into different species at Miocene (23 MYA to 5.33 MYA) [41]. Additionally, Takahashi has found an otolith fossil of *Sillago* from Niigata Prefecture (Japan) that may have existed during the Pliocene (5.3 MYA to 2.58 MYA) [42]. In the present study, the differentiation times based on amino acid sequences and variation sites are consistent with those in previous studies, although there existed a significant bias when using nucleotide sequences. We suspect that the base compositional biases caused by high-throughput sequencing affects the subsequent differentiation time analysis. However, it is undeniable that the tolerance of amino acid sequences to degenerate bases can effectively reduce this deviation [37]. Future studies still need to verify whether amino acid sequences and variation sites are more suitable for estimating the evolutionary order of species. Surprisingly, we used branch length to predict the differentiation times with other Perciformes species that might be quite different from those in the time tree database. There are two probable reasons: (1) the divergence in differentiation time results may be influenced by the estimation strategies and the number of genes used; (2) the orthologous exon markers obtained from transcriptome data are mostly functional genes, and the convergent evolution of functional genes has an inevitable effect on the evaluation of the species differentiation time. 

In brief, transcriptome data can provide hundreds to thousands of single-copy orthologous exon markers, and then be used to reconstruct a more complete view of *Sillago* species phylogenetic relationships. However, when using orthologous exon markers obtained from transcriptome data to reconstruct phylogenetic relationships, two recommendations are worth considering: (1) the amino acid sequences of orthologous exon markers can reduce errors caused by base compositional biases, so it is necessary for phylogenetic relationship reconstruction; (2) the accuracy of phylogenetic relationships may be positively correlated with the number of orthologous exon markers used, which is possibly because a large number of markers can eliminate the bias from a small number of convergent evolutionary genes. 

### 4.3. Positively selected genes Might Contribute to the Ecological Adaptation of Sillago Species

Transcriptome-wide analysis of the rates of non-synonymous to synonymous orthologous nucleotide substitutions represents an effective approach to quantitatively measure the selection force [17,43]. To reveal the molecular mechanism underlying the ecological adaptation of *Sillago* species, we estimated the dN/dS to identify the PSGs of *Sillago* species. A total of 44 orthologous genes were identified to be positively selected and might be involved in many biological processes, including the stress response (*LTV1* [44], *SMO* [45], *PSA* [46,47], *ABCB7* [48], *UBIAD1* [49], *COPA* [50], *MED27* [51,52], *MED28* [51,52], *SF3A1* [53], *SF3B5* [53], *TFIP11* [54], *NUDT6* [55], *POLλ* [56], and *DEPDC5* [57]), energy metabolism (*APF* [58] and *IMMT* [59]), carbohydrate metabolism (*SUCLG1* [60]), amino acid metabolism (*GCN4* [61] and *CPD* [62]), lipid metabolism (*HUWE1* [63] and *FABZ* [64]), visual sense (*AP4B1* [65] and *PRPF8* [66]), growth and differentiation (*ABTB1* [67], *UBE4A* [68], *DOHH* [69], and *GAB1* [70]), embryogenesis (*SBDS* [71], *TAF5L* [72], and *SEC8* [73]), and others. 

We suspected that the complexity (i.e., localized oxygen depletion, sulfide accumulation, and high turbidity) of the habitat environment may make *Sillago* species subject to multiple environmental stressors. Multiple environmental stressors might contribute to DNA damage and immunosuppression in *Sillago* species [74,75]. Meanwhile, the burrowing behavior of *Sillago* species may cause mechanical injuries to skin, and thus pathogens may enter the organism from the wound. Therefore, we suspected that those PSGs related to the stress response reflect the plasticity of *Sillago* species’ adaptation to multiple environmental stressors. A previous study has considered that the foraging probability and food-intake of *Sillago* species larvae may be limited by low light [76]. The inadequate intake can further affect all kinds of behaviors that necessitate high energy consumption, such as predation, reproduction, and others [77]. Therefore, it is a questionable whether the positive selection of material and metabolism-related genes related to material and energy metabolism could maintain the energy compensation of *Sillago* species. Interestingly, two vision genes were found to be positively selected in *Sillago* species. The positive selection of vision genes can enhance the visual acuity of *Sillago* species, which is useful for some behaviors (predation, reproduction, and others) in low light [78]. Meanwhile, the retina of *Sillago* species may have possessed a well-developed vascularization due to the positive selection of vision genes, possibly to overcome the hypoxic conditions [78]. A previous study has also considered that light can affect the foraging, growth, and reproductive behavior and the circadian rhythms of fishes [79]. It has been discussed above that multiple environmental stressors may affect the stress response, predation behavior, material and energy metabolism, and visual sensitivity of *Sillago* species, which may eventually limit the survival and development of *Sillago* species. Therefore, we suspected that *Sillago* species might positively select a battery of genes associated with growth, differentiation, and embryogenesis to maintain their effective population numbers under multiple environmental stressors. 

All in all, these PSGs might contribute to the ecological adaptation of *Sillago* species to the multiple environmental stressors, and these PSGs are also crucial to the evolution of *Sillago* species. However, our current evidence only shows specific protein sequence mutation in *Sillago* species. Whether these PSGs lead to favorable mutations in the phenotype of fish is unknown. Meanwhile, future experiments are also needed to explore which ecological traits of fish might have evolved along with these PSGs.

## 5. Conclusions

This study is the first systematic report of the transcriptome resource of *Sillago* species, and these data enrich the genomic information for molecular studies of these species. Based on eight well-annotated model fish genomes, we obtained 140 orthologous exon markers shared in *Sillago* species and then reconstructed a more complete phylogenetic relationship. Through the rates of non-synonymous to synonymous orthologous nucleotide substitutions, we found positive selection traces in 44 genes, and these genes are potentially related to the stress response, material (carbohydrate, amino acid, and lipid) and energy metabolism, growth and differentiation, embryogenesis, visual sense, and other things. This suggests that multiple environmental stressors may have led to specific selection force towards key genes in *Sillago* species. However, further experiments are needed to determine the exact function of these PSGs in *Sillago* species. Taken together, our results reconstruct a more complete view of the phylogenetic relationships of *Sillago* species based on transcriptome resources. We also suspect that the capacity of *Sillago* species to thrive in multiple environmental stressors may be due to the positive selection of these adaptive genes. The present study only represents a first step in understanding the habitat adaptive mechanism of the fish family Sillaginidae at the molecular level; further studies are still needed to validate the results and hypotheses.

## Figures and Tables

**Figure 1 animals-10-00633-f001:**
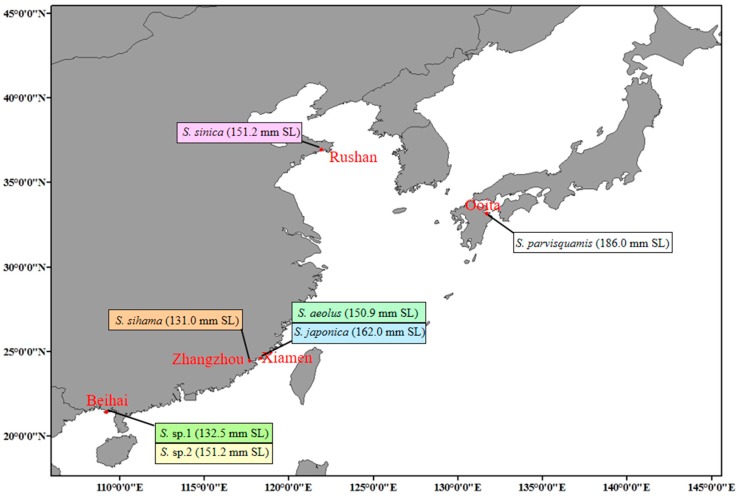
The sampling location and standard length (SL) of each sequencing individual.

**Figure 2 animals-10-00633-f002:**
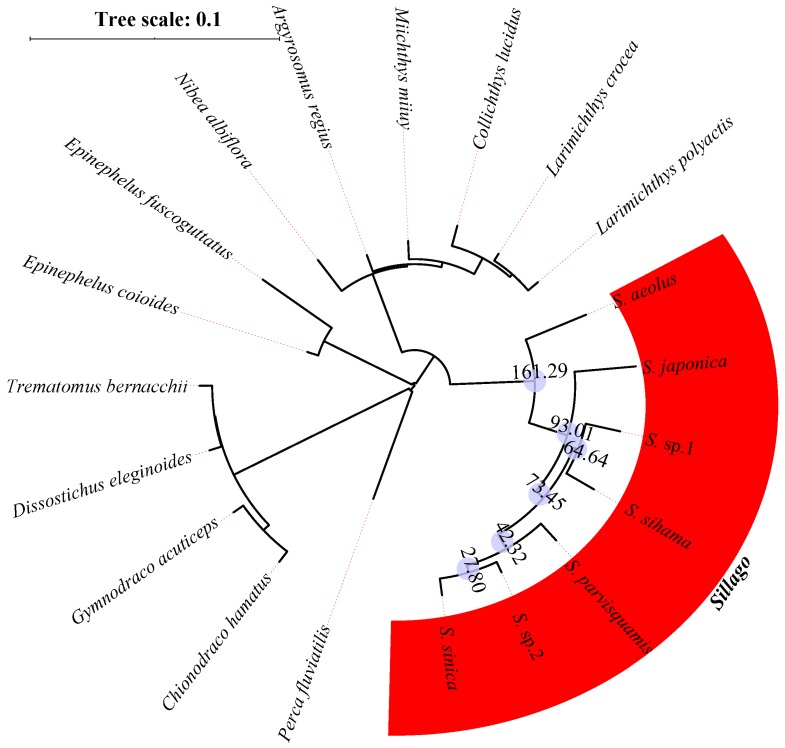
Inferred phylogenetic relationships and divergence times (data in the blue circles) of the seven *Sillago* species based on the concatenated nucleotide sequences.

**Figure 3 animals-10-00633-f003:**
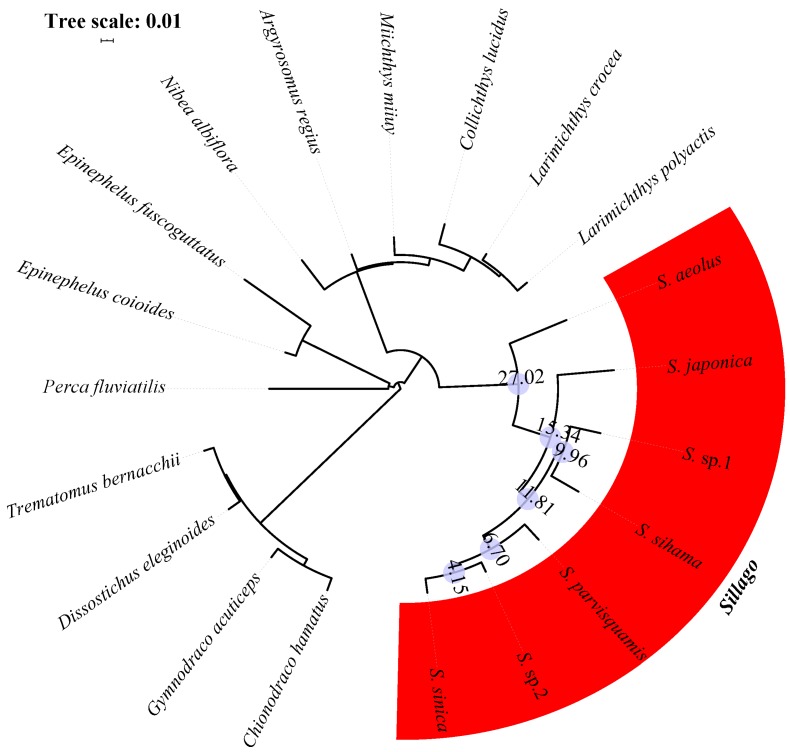
Inferred phylogenetic relationships and divergence times (data in the blue circles) of the seven *Sillago* species based on the concatenated amino acid sequences.

**Figure 4 animals-10-00633-f004:**
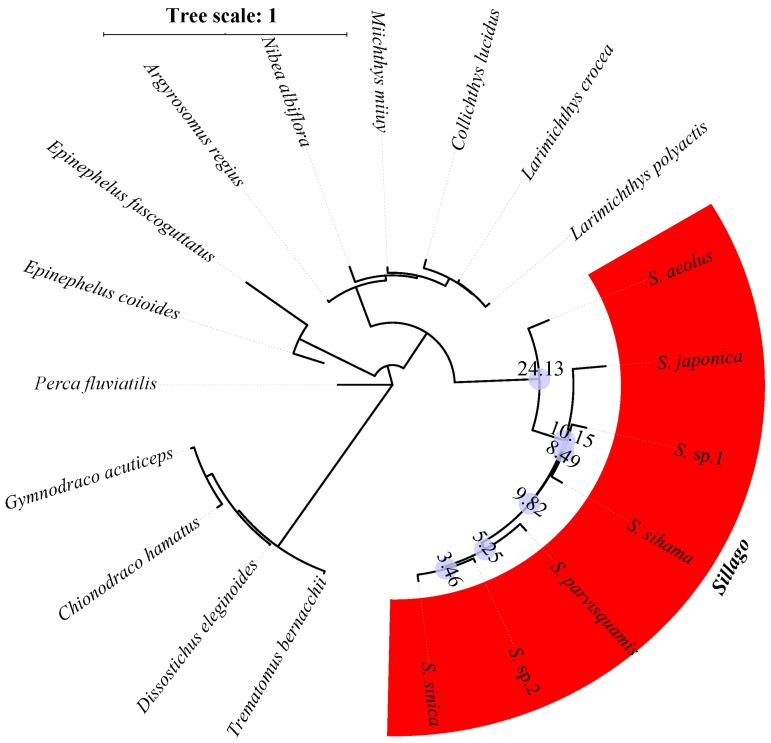
Inferred phylogenetic relationships and divergence times (data in the blue circles) of the seven *Sillago* species based on the concatenated variation sites.

**Figure 5 animals-10-00633-f005:**
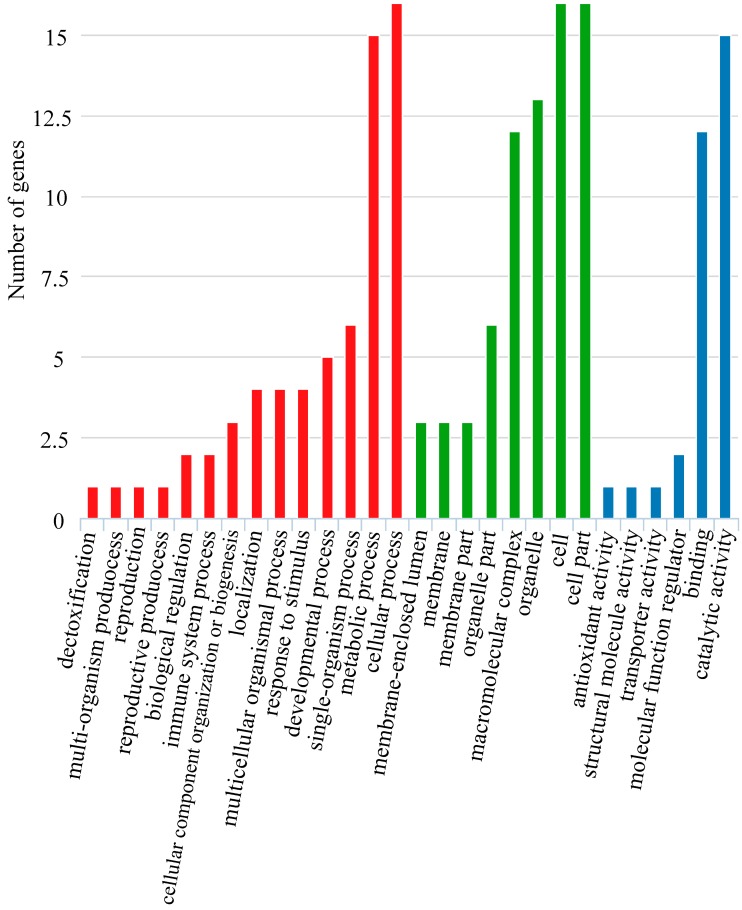
GO enrichment analysis of representative positively selected genes.

**Table 1 animals-10-00633-t001:** The ecological characteristics of 20 research species.

Species	Classification	Milieu	Climate Zone	Depth Range (M)	Maturity Length (cm)	Feeding Habits	Type of Fish Eggs
*E. fuscoguttatus*	Serranidae	Marine; brackish; reef-associated	Tropical	1–60	50	Carnivorous	Pelagic
*E. coioides*	Serranidae	Marine; brackish; reef-associated	Subtropical	1–100	25–30	Carnivorous	Pelagic
*P. fluviatilis*	Percidae	Freshwater; brackish; demersal	Temperate	1–30	11–23.4	Carnivorous	Adhesive
*C. hamatus*	Channichthyidae	Marine; demersal	Polar	4–600	33–37	Carnivorous	Pelagic
*G. acuticeps*	Bathydraconidae	Marine; demersal	Polar	0–550	-	Carnivorous	Pelagic
*D. eleginoides*	Nototheniidae	Marine; demersal	Temperate	50–3850	38–60	Carnivorous	Pelagic
*T. bernacchii*	Nototheniidae	Marine; demersal;	Polar	0–700	18	Carnivorous	Pelagic
*A. regius*	Sciaenidae	Marine; brackish; demersal	Subtropical	15–300	80	Carnivorous	Pelagic
*N. albiflora*	Sciaenidae	Marine; demersal; coastal waters with mudddy to sanddy-muddy bottoms	Temperate	25–80	-	Carnivorous	Pelagic
*M. miiuy*	Sciaenidae	Marine; brackish; demersal; coastal waters with mudddy to sanddy-muddy bottoms	Temperate	15–100	-	Carnivorous	Pelagic
*C. lucidus*	Sciaenidae	Marine; demersal; coastal waters with mudddy to sanddy-muddy bottoms	Subtropical	0–90	13	Carnivorous	Pelagic
*L. polyactis*	Sciaenidae	Marine; demersal; sublittoral zone above 120 m with muddy to sanddy-muddy bottoms	Subtropical	0–120	18.1	Carnivorous	Pelagic
*L. crocea*	Sciaenidae	Marine; brackish; demersal; coastal waters and estuaries with muddy to muddy-sandy bottoms shallower than 120 m depth	Temperate	0–120	17	Carnivorous	Pelagic
*S. aeolus*	Sillaginidae	Marine; demersal; nearshore shallow and estuarine waters; burrowing life-style	Tropical	0–60	12	Carnivorous	Pelagic
*S. japonica*	Sillaginidae	Marine; demersal; nearshore shallow and estuarine waters; burrowing life-style	Subtropical	0–30	-	Carnivorous	Pelagic
*S. parvisquamis*	Sillaginidae	Marine; brackish; demersal; nearshore shallow and estuarine waters; burrowing life-style	Subtropical	0–30	-	Carnivorous	Pelagic
*S. sihama*	Sillaginidae	Marine; brackish; reef-associated; nearshore shallow and estuarine waters; burrowing life-style	Tropical	0–60	13–19.1	Carnivorous	Pelagic
*S. sinica*	Sillaginidae	Marine; brackish; demersal; nearshore shallow and estuarine waters; burrowing life-style	Tropical	-	-	Carnivorous	Pelagic
*S.* sp.1	Sillaginidae	-	-	-	-	-	-
*S.* sp.2	Sillaginidae	-	-	-	-	-	-

Note: “-” indicates that no statistics were found.

**Table 2 animals-10-00633-t002:** The clean transcriptomic reads of the seven *Sillago* species.

*Sillago* Species	Read Number	GC%	%≥Q30
*S. aeolus*	78,709,246	51.14	92.37
*S. japonica*	50,013,641	53.02	92.96
*S. parvisquamis*	113,351,008	52.88	93.75
*S. sihama*	87,050,702	51.34	92.63
*S. sinica*	97,977,199	52.19	94.51
*S.* sp.1	51,710,081	53.86	93.74
*S.* sp.2	70,996,526	53.57	92.95

**Table 3 animals-10-00633-t003:** The transcriptome assembly information of the seven *Sillago* species.

*Sillago* Species	Unigene			
	Number	Total Length (bp)	Mean Length (bp)	N50 Length (bp)
*S. aeolus*	82,024	51,896,226	787.32	1,403
*S. japonica*	58,102	23,966,004	428.99	461
*S. parvisquamis*	102,185	79,280,211	1,019.38	1,986
*S. sihama*	69,748	48,391,713	815.81	1,369
*S. sinica*	102,903	78,264,349	992.70	1,848
*S.* sp.1	63,807	34,524,368	588.49	738
*S.* sp.2	85,990	49,751,159	652.68	902

**Table 4 animals-10-00633-t004:** Representative positively selected genes in *Sillago* species.

	Gene Name	Description	×10-Value	FDR-Adjusted *p*-Value
Stress response	*MED27*	mediator of RNA polymerase II transcription subunit 27	3.02 × 10^−39^	0.00
	*MED28*	mediator of RNA polymerase II transcription subunit 28	2.33 × 10^−29^	0.00
	*LTV1*	protein LTV1 homolog	1.08 × 10^−37^	7.89 × 10^−03^
	*SMO*	Spermine oxidase	2.21 × 10^−22^	1.27 × 10^−14^
	*PSA*	puromycin-sensitive aminopeptidase	3.70 × 10^−41^	0.00
	*ABCB7*	ATP-binding cassette sub-family B member 7, mitochondrial	5.54 × 10^−31^	0.00
	*COPA*	coatomer subunit alpha	4.99 × 10^−45^	0.00
	*SF3A1*	splicing factor 3A subunit 1	2.31 × 10^−44^	0.00
	*SF3B5*	splicing factor 3B subunit 5	8.30 × 10^−60^	0.00
	*DEPDC5*	GATOR complex protein DEPDC5 isoform X3	1.72 × 10^−46^	0.00
	*POLλ*	DNA polymerase lambda	1.48 × 10^−80^	0.00
	*TFIP11*	tuftelin-interacting protein 11	2.10 × 10^−72^	0.00
	*NUDT6*	Nucleoside diphosphate-linked moiety X motif 6	1.32 × 10^−83^	0.00
	*UBIAD1*	UbiA prenyltransferase domain-containing protein 1	7.59 × 10^−96^	0.00
Energy metabolism	*APF*	bis(5′-nucleosyl)-tetraphosphatase [asymmetrical]	5.30 × 10^−64^	0.00
	*IMMT*	MICOS complex subunit MIC60 isoform X2	1.66 × 10^−126^	1.38 × 10^−04^
Carbohydrate metabolism	*SUCLG1*	succinate-CoA ligase [ADP/GDP−forming] subunit alpha, mitochondrial	2.37 × 10^−23^	0.00
Amino acid metabolism	*GCN1*	eIF-2-alpha kinase activator GCN1	1.82 × 10^−41^	0.00
	*CPD*	Carboxypeptidase D	6.74 × 10^−46^	0.00
Lipid metabolism	*HUWE1*	E3 ubiquitin-protein ligase HUWE1 isoform X1	3.95 × 10^−35^	1.15 × 10^−03^
	*HUWE1*	E3 ubiquitin-protein ligase HUWE1 isoform X1	4.55 × 10^−24^	0.00
	*HUWE1*	E3 ubiquitin-protein ligase HUWE1 isoform X1	8.49 × 10^−54^	7.85 × 10^−03^
	*HUWE1*	E3 ubiquitin-protein ligase HUWE1 isoform X1	1.31 × 10^−38^	0.00
	*HUWE1*	E3 ubiquitin-protein ligase HUWE1 isoform X1	1.48 × 10^−40^	0.00
	*HUWE1*	E3 ubiquitin-protein ligase HUWE1 isoform X1	1.40 × 10^−41^	0.00
	*FABZ*	hydroxyacyl-thioester dehydratase type 2, mitochondrial	7.36 × 10^−80^	0.00
	*HUWE1*	E3 ubiquitin-protein ligase HUWE1 isoform X1	2.11 × 10^−117^	5.07 × 10^−03^
	*HUWE1*	E3 ubiquitin-protein ligase HUWE1 isoform X1	1.57 × 10^−117^	0.00
Visual sense	*AP4B1*	AP-4 complex subunit beta-1	5.00 × 10^−45^	0.00
	*PRF8*	Pre-mRNA-processing-splicing factor 8	2.12 × 10^−52^	0.00
Growth and differentiation	*ABTB1*	ankyrin repeat and BTB/POZ domain-containing protein 1	1.96 × 10^−36^	0.00
	*UBE4A*	ubiquitin conjugation factor E4 B isoform X2	1.31 × 10^−29^	1.11 × 10^−12^
	*GAB1*	GRB2-associated-binding protein 1 isoform X1	3.65 × 10^−61^	0.00
	*DOHH*	deoxyhypusine hydroxylase	2.66 × 10^−64^	0.00
Embryogenesis	*SBDS*	ribosome maturation protein SBDS	2.55 × 10^−39^	0.00
	*SEC8*	exocyst complex component 8	1.38 × 10^−90^	0.00
	*TAF5L*	TAF5-like RNA polymerase II p300/CBP-associated factor-associated factor 65 kDa subunit 5L	0.00	0.00
Others	*CCDC25*	Coiled-coil domain-containing protein 25	9.63 × 10^−17^	0.00
	*PIGY*	phosphatidylinositol N-acetylglucosaminyltransferase subunit Y	1.12 × 10^−40^	0.00
	*-*	fumarylacetoacetate hydrolase domain-containing protein 2-like isoform X2	6.37 × 10^−43^	0.00
	*USP24*	ubiquitin carboxyl-terminal hydrolase 24 isoform X2	5.15 × 10^−31^	0.00
	*RPN2*	26S proteasome non-ATPase regulatory subunit 1	2.38 × 10^−44^	0.00
	*CXORF56*	UPF0428 protein CXorf56 homolog	1.53 × 10^−136^	0.00
	*TALIN*	talin rod domain-containing protein 1	0.00	3.81 × 10^−06^

**Table 5 animals-10-00633-t005:** KEGG pathway enrichment analysis of representative PSGs.

Pathway	Pathway_ID	Key Enzyme	Gene Name
Nicotinate and nicotinamide metabolism	map00760	diphosphatase	*APF*
Carbon fixation pathways in prokaryotes	map00720	ligase (ADP-forming)	*SUCLG1*
Purine metabolism	map00230	adenylpyrophosphatase; diphosphatase; phosphatase	*ABCB7*, *APF*, *ABCB7*
C5-Branched dibasic acid metabolism	map00660	ligase (ADP-forming)	*SUCLG1*
Starch and sucrose metabolism	map00500	diphosphatase	*APF*
Arginine biosynthesis	map00220	synthase (NADPH)	*UBIAD1*
Riboflavin metabolism	map00740	diphosphatase	*APF*
Pantothenate and CoA biosynthesis	map00770	diphosphatase	*APF*
Pyrimidine metabolism	map00240	diphosphatase	*APF*
Biosynthesis of antibiotics	map01130	synthase (NADPH); ligase (ADP-forming); ligase (GDP-forming)	*UBIAD1*, *SUCLG1*, *SUCLG1*
Citrate cycle (TCA cycle)	map00020	ligase (ADP-forming); ligase (GDP-forming)	*SUCLG1*, *SUCLG1*
Propanoate metabolism	map00640	ligase (ADP-forming); ligase (GDP-forming)	*SUCLG1*, *SUCLG1*
Thiamine metabolism	map00730	Phosphatase	*ABCB7*
Arginine and proline metabolism	map00330	synthase (NADPH)	*UBIAD1*

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
