# Peer review of "Comprehensive Transcriptome Analysis Reveals Insights into Phylogeny and Positively Selected Genes of Sillago Species"

_animals, 2020, doi:10.3390/ani10040633_

Round 1
Reviewer 1 Report
I confirmed that the manuscript is revised in response to reviewer's comments. Answers from authors are also reasonable and conscientious. I agree to be published as is.
Author Response
Thanks very much for your comments.
Reviewer 2 Report
I think this manuscript is worth publishing in Animals without revision.
I expect authors to compare the sequence variability of the intron of PSGs within and between species to confirm the existence of on-going selection.
Author Response
Thanks very much for your comments.
I expect authors to compare the sequence variability of the intron of PSGs within and between species to confirm the existence of on-going selection.
Re: I agree with the view that " compare the sequence variability of the intron of PSGs within and between species to confirm the existence of on-going selection". In the present study, we only confirm the specific evolution of Sillago species. In the future, we will follow your suggest and study the specific evolution of a particular ecological trait of an interested Sillago species.
This manuscript is a resubmission of an earlier submission. The following is a list of the peer review reports and author responses from that submission.
Round 1
Reviewer 1 Report
The article is well-written and includes important suggestive results about divergence and adaptive process in Sillago species. So, it is worthy to be published. However, I request some revision in response to my comments.
Comments,
- Samples of Sillago
The precise location, the number of each sample should be described. I have a question why the muscle only was analyzed. It seems insufficient to speculate the adaptation to environment.
- Characteristics of research species.
I recommend making the table to show the characteristics of research species to make clear the ecological differences between research species and Sillago species.
- Phylogenetic relationship
Is there any possibility that the relationship within the seven Sillago (Figure1.) includes interspecies or geological variation?
Author Response
Thanks very much for the suggestions from reviewer 1 on our manuscript. These comments helped us to improve the manuscript. The following are our response to the reviewer 1 comments.Reviewer 11. Samples of SillagoThe precise location, the number of each sample should be described. I have a question why the muscle only was analyzed. It seems insufficient to speculate the adaptation to environment.
Re: According to reviewer-1’s comment, we have provided the information of location and number of each sample. It's worth noting that we took a large number of samples for each species, but only one sample was used for transcriptome sequencing.Muscle tissue is the largest energy and amino acid pool in the maintenance of homeostasis, and it can well demonstrates the effect of environmental changes on marine organisms (Logan and Buckley, 2015). Therefore, we selected muscle tissue for transcriptome analysis. Additionally, singly-copy orthologous genes have been applied to investigate phylogenetic relationships and adaptive evolutionary mechanisms of Sillago species. Single-copy orthologous genes refers to the single copy and formed housekeeping genes in the genome, which are very conservative and inherited directly in the biological evolution (Duarte et al., 2010; Creevey et al., 2011). We suspected that single-copy orthologous genes are conserved in all tissues, but their expression levels are tissue-specific. Additionally, a single-copy conserved nuclear coding sequences database of eight model fishes was constructed and used to search for homologous sequences for 20 research species. This method can search for relatively complete single-copy orthologous genes for subsequent analysis. Therefore, only muscle tissue was selected in this study. However, it is undeniable that the sequencing of more tissues can improve the integrity of transcriptome information, and we will also use more tissues for transcriptome sequencing according to the opinions of reviewers in subsequent studies. We hope our explanation will give you satisfaction.Reference:Logan, C.A.; Buckley, B.A. Transcriptomic responses to environmental temperature in eurythermal and stenothermal fishes. J. Exp. Biol. 2015, 218, 1915-1924. Duarte, J. M.; Wall, P.K.; Edger, P.; Landherr, L.; Ma, H.; Pires, J.; Mack, J.; dePamphilis, C. Identification of shared single copy nuclear genes in Arabidopsis, Populus, Vitis and Oryza and their phylogenetic utility across various taxonomic levels. BMC Evol. Biol. 2010, 10: 61.Creevey, C. J.; Muller, J.; Doerks, T.; Thompson, J.D.; Arendt, D.; Bork, P. Identifying Single Copy Orthologs in Metazoa. Plos Comput. Biol. 2011, 7(12): e1002269.
2. Characteristics of research species.I recommend making the table to show the characteristics of research species to make clear the ecological differences between research species and Sillago species.
Re: According to reviewer’s comment, we made a detailed comparison of the ecological characteristics between the Sillago species and outgroups, and the results were presented in a tabular form. Based on the comparison results of ecological characteristics, we are rediscovering that our research is described the adaptive evolution of whitings adaptations to complex benthic environments, not only adaptive evolution to the benthic sandy environment. Therefore, we changed the description and we have replaced “benthic sandy environmental adaptation” with “benthic environmental adaptation” in our manuscript. I believe that the benthic environment with more environmental factors can affect multiple ecological characteristics of Sillago species, thus the description of benthic environment may be more accurate. We hope our explanation will give you satisfaction.
3.Phylogenetic relationshipIs there any possibility that the relationship within the seven Sillago (Figure1.) includes interspecies or geological variation?
Re: We suspect that the description of the “suspected species” may have been used in our manuscript, which ultimately led the reviewer to question whether the S.sp.1 and S.sp.2 might have originated from the differences between different populations. In fact, our previous research confirms that there are 10 effective Sillago species off the coast of mainland China, including S. chondropus, S. sihama, S. sinica, S. shaoi, S. aeolus, S. asiatica, S. ingenuua, S. japonica, and two unpublished Sillago species (S.sp.1 and S.sp.2) mentioned in this study. In fact, we have determined two species (S.sp.1 and S.sp.2) at the morphological, anatomical, and molecular levels and found that they are different, but unfortunately the results have not yet been published. In order to eliminate the trouble caused by our description, the two unpublished species was briefly explained in the introduction. The new description as follows: “five valid Sillago species (including S. japonica, S. aeolus, S. sihama, S. parvisquamis, and S. sinica) and two unpublished new species (S. sp.1 and S. sp.2), which were confirmed by using morphological, anatomical and DNA-barcoding evidence, were sampled from the coast of China”. We hope our explanation will give you satisfaction.
Reviewer 2 Report
The aim is clear, the data is novel and the finding will be useful for the understanding of the evolution of the Sillago species, this report is basically worth publishing in Animals after revising following points:
1) I think authors should provide the detailed description about the samples, such as, sampling location, body size, sex.
2) I recommend authors to assess the completeness of the transcriptome data by BUSCO. Besides, it will be better to show the gene expression patterns of the samples revealed by the trancriptome analysis.
Author Response
Thanks very much for the suggestions from reviewer 2 on our manuscript. These comments helped us to improve the manuscript. The following are our response to the reviewer 2 comments.
Reviewer 2
1. I think authors should provide the detailed description about the samples, such as, sampling location, body size, sex.
Re: According to reviewer comment, we have provided the detailed description about the sampling location, body size, sex. And the information of sampling location and the average body length range was showed in Figure. It's worth noting that we took a large number of samples for each species, but only one sample was used for transcriptome sequencing. Additionally, only female samples were used for sequencing to eliminate gender-based differences in transcriptome information.
2. I recommend authors to assess the completeness of the transcriptome data by BUSCO. Besides, it will be better to show the gene expression patterns of the samples revealed by the trancriptome analysis.
Re: According to reviewer comment, we have assessed the completeness of the transcriptome data by BUSCO. Additionally, I strongly agree with the reviewer's suggestion on the analysis of gene expression patterns. The analysis of gene expression patterns was beneficial to verify whether the 44 positive genes could improve the adaptability of Sillago species to complex environments. However, only the data of Sillago species in this study were obtained by transcriptome sequencing, while the data of other research species were all obtained by genome sequencing, thus limiting the analysis of gene expression patterns. In my further study, we will obtain the transcriptome data of other research species and analyze gene expression patterns by the trancriptome analysis.